# Reference Gene Validation for RT–qPCR in PBMCs from Asthmatic Patients with or without Obesity

**DOI:** 10.3390/mps5030035

**Published:** 2022-04-22

**Authors:** Marina Bantulà, Ebymar Arismendi, César Picado, Joaquim Mullol, Jordi Roca-Ferrer, Valeria Tubita

**Affiliations:** 1August Pi i Sunyer Biomedical Research Institute (IDIBAPS), Hospital Clinic, Universitat de Barcelona, 08036 Barcelona, Spain; earismen@clinic.cat (E.A.); cpicado@ub.edu (C.P.); jmullol@clinic.cat (J.M.); jrocaf@clinic.cat (J.R.-F.); v.tubita@gmail.com (V.T.); 2Department of Pneumology, Hospital Clinic, 08036 Barcelona, Spain; 3Centre for Biomedical Investigation in Respiratory Diseases (CIBERES), 08036 Barcelona, Spain; 4Rhinology Unit and Smell Clinic, ENT Department, Hospital Clinic, 08036 Barcelona, Spain

**Keywords:** reference genes, RT–qPCR, obesity, asthma, BestKeeper, ΔCt, geNorm, NormFinder

## Abstract

Obesity is known to impair the efficacy of glucocorticoid medications for asthma control. Glucocorticoid-induced gene expression studies may be useful to discriminate those obese asthmatic patients who present a poor response to glucocorticoids. The expression of genes of interest is normalized with respect to reference genes (RGs). Ideally, RGs have a stable expression in different samples and are not affected by experimental conditions. The objective of this work was to analyze suitable RGs to study the role of glucocorticoid-induced genes in obese asthmatic patients in further research. The gene expression of eight potential RGs (*GUSB*, *B2M*, *POLR2A*, *PPIA*, *ACTB*, *GAPDH*, *HPRT1*, and *TBP*) was assessed with reverse transcription–quantitative polymerase chain reaction in peripheral blood mononuclear cells (PBMCs) from asthmatic, obese asthmatic, and healthy individuals. Their stability was analyzed using four different algorithms—BestKeeper, ΔCt, geNorm, and NormFinder. geNorm analysis recommended the use of a minimum of three genes for normalization. Moreover, intergroup variation due to the treatment was calculated by NormFinder, which found that *B2M* was the gene that was least affected by different treatments. Comprehensive rankings indicated *GUSB* and *HPRT1* as the best RGs for qPCR in PBMCs from healthy and asthmatic subjects, while *B2M* and *PPIA* were the best for obese asthmatic subjects. Finally, our results demonstrated that *B2M* and *HPRT1* were the most stable RGs among all groups, whereas *ACTB*, *TBP*, and *GAPDH* were the worst shared ones.

## 1. Introduction

Synthetic glucocorticoids (GCs) are a large family of anti-inflammatory agents used in asthma as the initial recommended controller treatment [1]. When asthma is associated with obesity (defined as having a body mass index of ≥30 kg/m^2^), patients present a decreased response to GC treatment [2] that results in poor asthma control with an increase in the frequency of exacerbations and a greater severity of respiratory symptoms [3,4]. Furthermore, obese patients present low levels of vitamin D, and it is known that this is associated with impaired lung function, higher asthma exacerbation frequency, and a reduced response to GCs [5,6,7].

Currently, there are no biomarkers to predict GC effectiveness; however, genetic testing may help discriminate patients with different sensitivities to GCs. The expression of anti-inflammatory genes induced by GCs in peripheral blood mononuclear cells (PBMCs) represents a useful in vitro method to predict and evaluate clinical responses to GCs. PBMCs are a good and long-lasting source of nucleic acids for this type of study [8]. Comparisons of their gene expression profiles in asthmatic subjects with and without obesity can provide information on the pathophysiological mechanisms underlying the poor response to GC in these patients.

Reverse transcription (RT)—quantitative polymerase chain reaction (qPCR) has become the method of choice for mRNA gene expression analysis [9,10] due to its high sensitivity, specificity, broad quantification range [11], low cost, and reliability [12]. However, the accuracy of the results may be influenced by several factors, including inter-sample variations, sample collection, RNA preparation and quality, and the efficiency of RT and qPCR amplification [13,14]. To mitigate differences in RNA sampling, the normalization of gene expression data against reference genes (RGs) is critical to avoid the misinterpretation of results in gene expression studies [15]. An ideal RG must be stably expressed, non-regulated, and unaffected by biological or experimental conditions [13,16]. However, numerous studies have shown that commonly used RGs considerably vary in different disease processes or different tissue and cell types [17,18]. For this reason, it is essential to validate the suitability of potential RGs for each experimental condition [19].

To date, no studies that have validated RGs in PBMCs treated with GCs from asthmatic patients with or without obesity compared to healthy subjects. Here, we aimed to analyze gene expression stability of eight commonly used RGs via RT–qPCR using four different algorithms—BestKeeper [20], ΔCt [21], geNorm [13], and NormFinder [14]—in order to validate their reliability for further gene expression studies involving PBMCs from obese asthmatic patients treated in vitro with dexamethasone (DEX) and/or the active form of vitamin D (1,25-dihydroxyvitamin D_3_) (Vit D_3_).

## 2. Materials and Methods

### 2.1. Patients and Control Subjects

The study was conducted with PBMCs from diagnosed asthmatic patients (*n* = 3) and obese asthmatic patients (*n* = 3) and with age- and gender-matched healthy individuals (*n* = 3). The patients’ clinical features are shown in Appendix A. Asthmatic and obese asthmatic patients were not under systemic corticosteroid use.

The study was conducted with informed written consent from the participating subjects. The study was approved by the ethics committee of our institution (HCB/2016/0861), and all methods were carried out in accordance with relevant guidelines and regulations.

### 2.2. Blood Collection and PBMCs Isolation

Twenty milliliters of whole blood was collected from each patient via venipuncture into a vacutainer tube containing an anticoagulant (EDTAK2). PBMCs were isolated using Lymphoprep^TM^ (Stem Cell TM) following the manufacturer’s instructions. Whole blood was diluted to 1:1 with a balanced salt solution, and this was then layered on top of the Lymphoprep solution in a 50 mL conical tube and centrifuged at 800× *g* for 20 min at room temperature. After centrifugation, the white ring containing mononuclear cells was transferred to a new tube and washed twice with balanced salt solution [22].

### 2.3. Cell Culture Conditions

PBMCs were cultured at 37 °C in a 5% CO_2_ atmosphere using an RPMI-1640 medium supplemented with 5% charcoal-stripped fetal bovine serum and antibiotics (100 U/mL of penicillin and 100 µg/mL streptomycin), with a final cell concentration of 2 × 10^6^/mL.

The cell suspension from each patient was divided into four wells according to the following experimental conditions: negative control well (cells with culture medium), DEX at a final concentration of 1 × 10^−^^5^ M, Vit D_3_ at a final concentration of 1 × 10^−^^7^ M, and DEX 1 × 10^−^^5^ M with Vit D_3_ 1 × 10^−^^7^ M together. These conditions were repeated in order to harvest cells at two different time-points: 6 and 24 h.

### 2.4. RNA Extraction and cDNA Synthesis

Total RNA was isolated from PBMCs using the TRIzol (Life Technologies, Paisley, UK) reagent according to the manufacturer’s protocol. Total mRNA concentration was measured at 260 nm, and purity was assessed from the 260/280 nm absorbance ratio. RNA was treated with DNase I (Thermo Fisher, Vilnius, Lithuania) for 30 min at 37 °C to remove any potential genomic DNA contamination. We converted 1 μg of RNA into cDNA using the High-Capacity cDNA Reverse Transcription Kit according to the manufacturer’s instructions (Thermo Fisher, Vilnius, Lithuania). Samples were incubated for 10 min at 25 °C, 120 min at 37 °C, and 5 min at 85 °C. Final cDNA products were diluted 10-fold prior to use in qPCR.

### 2.5. Real-Time qPCR

The expression of eight selected RGs (Appendix A) was analyzed with real-time qPCR. qPCR experiments were carried out using 100 ng of cDNA in the Viia7 Real-Time PCR system (Applied Biosystems, Carlsbad, CA, USA) following the manufacturer’s guidelines. The qPCR reaction consisted of 0.5 µL of TaqMan gene Expression Assay (20X), 5 µL of TaqMan Universal PCR Master Mix (2X), 2.5 µL of RNase-free water, and 2 µL of cDNA in a total volume of 10 µL. The thermal cycler was set to 95 °C for 20 min, followed by 40 reaction cycles of 1 s at 95 °C and 20 s at 60 °C. All qPCR reactions were set up in duplicate.

### 2.6. Statistical Analysis

Gene expression stability was analyzed using four different, statistically-based methods: BestKeeper, ΔCt, geNorm, and NormFinder. Afterwards, a final ranking based on the results from these algorithms was calculated. 

Clinical data were reported as mean ± SD for normally distributed data and median (range) for experimental nonparametric data. Comparisons of gene expression levels between subgroups were performed using the Kruskal–Wallis H test for nonparametric data followed by post hoc Dunn’s multiple comparisons test. The Wilcoxon test was used to compare paired samples. All analyses were performed using GraphPad Prism version 6.02 for Windows, (GraphPad Software, La Jolla, CA, USA). Statistical significance was defined as *p*-value < 0.05.

## 3. Results

### 3.1. Quantification Cycle (Cq) Characterization of Candidate RGs

Our results showed a wide range of expression levels (Appendix A). One of the samples showed a 3-fold intrinsic variance compared to the mean of all RGs, so this sample was excluded [20]. The final analysis contained 24 samples from healthy patients, 24 from asthmatic patients, and 23 from obese asthmatic patients. The high-abundance genes (Cq < 25) were *B2M* < *PPIA* < *ACTB* < *GAPDH* in healthy and obese asthmatic PBMCs and *B2M* < *ACTB* < *PPIA* < *GAPDH* in asthmatic PBMCs. The low-abundance genes (Cq > 25) were *HPRT1* > *TBP* > *GUSB* > *POLR2A* in the three groups studied.

### 3.2. Influence of Experimental Conditions 

First of all, we compared gene expression between 6 and 24 h of incubation time and found no differences in PBMCs from healthy individuals. However, *GAPDH* showed an increased expression after 24 h of incubation in both the asthmatic and obese asthmatic groups (*p* < 0.05 and *p* < 0.01, respectively). Moreover, *PPIA* and *TBP* presented differences in asthmatic PBMCs between 6 and 24 h (*p* < 0.05 and *p* < 0.01, respectively).

Secondly, we assessed the impact of in vitro treatments: DEX with or without the addition of Vit D_3_ for 6 and 24 h. In PBMCs from healthy individuals, there were no differences between experimental conditions; however, we found differences in gene expression in asthmatic and obese asthmatic PBMCs. DEX was the treatment that most affected stability: *POLR2A* and *GAPDH* gene expression presented differences at both 6 h (*p* < 0.05 and *p* < 0.01, respectively) and 24 h (*p* < 0.01 and *p* < 0.05, respectively). *HPRT1* and *TBP* showed differences after 6 h of incubation (*p* < 0.01 for both). Vit D_3_ treatment only affected *POLR2A* gene expression after 6 h of incubation (*p* < 0.05). No differences were found with Vit D_3_ at 24 h or with the co-incubation of DEX and Vit D_3_ at 6 h. *GUSB* and *POLR2A* showed differences with the combined treatment of DEX and Vit D_3_ after 24 h of incubation (*p* < 0.01 for both).

### 3.3. Analyses of Gene Expression Stability

To determine the stability values for each candidate RG, we used four different programs (BestKeeper, ΔCt, geNorm, and NormFinder).

#### 3.3.1. BestKeeper Analysis

BestKeeper determines the optimal RGs by performing a pairwise correlation approach and then ranking the RG candidates’ stability based on the standard deviation (SD) of Cq values; RGs with SD > 1 are excluded [20]. All candidate RGs showed a strong correlation (0.69 < r < 0.92) and were combined into an index. The highest Pearson correlation coefficient (r) value for the relationship between the index and the contributing RGs was obtained for *HPRT1* (r = 0.89, *p* = 0.001) in healthy subjects. In obese asthmatic patients, *HPRT1* also presented the highest correlation coefficient (r = 0.92, *p* = 0.001), whereas the highest coefficient in asthmatic patients was for *GAPDH* (r = 0.92, *p* = 0.001). SD was <1 for all genes when studying groups separately, signifying that all candidate RGs could be considered stable genes. The worst ranked gene was *GAPDH* for all three groups.

#### 3.3.2. ΔCt Analysis

ΔCt generates ‘pair of genes’ comparisons between each gene and the other RGs within each sample, and then it calculates the average SD against the other RGs [21]. All samples presented good stability values (SD < 1), but the samples that had the lowest SDs were *HPRT1* in healthy PBMCs, *GUSB* in asthmatic PBMCs, and *B2M* in obese asthmatic PBMCs.

#### 3.3.3. geNorm Analysis

geNorm provides a measure of gene expression stability (M) by calculating the average pairwise variation of each RG from all the other RG candidates. A low M value represents stable gene expression [13]. All studied genes had an M value below the default limit of 1.5, demonstrating that all tested genes had high expression stability. Genes with the lowest M value for each group were the same as for the ΔCt method. Furthermore, the least stable RGs also coincided between both algorithms, with *GAPDH* showing the lowest stability value for both healthy and obese asthmatic PBMCs and *B2M* showing the lowest stability value for asthmatic PBMCs.

Additionally, the optimal number of RGs needed to quantify the expression of the gene of interest was also determined using geNorm. Here, the pairwise variation (Vn/n+1) between the two sequential normalization factors is calculated, and when the V value is below the 0.15 cut-off value, it is not necessary to include additional RGs [13]. The V3/4 value was found to be below the threshold of 0.15 in healthy and obese asthmatic samples, indicating that three genes were sufficient for normalization. However, in asthmatic PBMCs, the optimal number of RGs for normalization was four. 

#### 3.3.4. NormFinder

NormFinder is an add-in for Microsoft Excel that uses a model-based approach to estimate intragroup and intergroup variation for each candidate RG. Both values are combined to give a stability value (S value), where the lowest value indicates the most stable expression [14]. The most stable gene in this study was *HPRT1*, and the ones with the highest S values were *GAPDH* for healthy and obese asthmatic PBMCs and *PPIA* for asthmatic PBMCs.

Moreover, intergroup variation was calculated by NormFinder to assess the influence of different cell culture treatments (Figure 1). The most stable genes with the least intergroup variation were *B2M* for healthy and obese asthmatic and *HPRT1* for asthmatic PBMCs.

### 3.4. Comprehensive Ranking

An overall final ranking was calculated for each RG as the geometric mean based on their weight according to the used statistical methods (Table 1). RGs were sorted from the most stable gene, having the lowest ranking position, to the least stable one, with the highest number. As shown in Table 1, in healthy and asthmatic PBMCs, *HPRT1* and *GUSB* were the best ranked RGs, whereas *B2M* and *HPRT1* were the most stable ones for the obese asthmatic group.

### 3.5. Identification of the Best and Worst Shared Scored Genes

Finally, we sought to identify the best and worst shared genes between healthy, asthmatic, and obese asthmatic groups. We selected the four best-scored genes and the four worst-scored genes from the comprehensive ranking for each group of patients, and we found the following coincidences: *B2M* and *HPRT1* were the best RGs among all groups, whereas *ACTB* and *TBP* were the worst shared RGs. The Venn diagram in Figure 2 illustrates these coincidences.

## 4. Discussion

The authors of the present study aimed to evaluate the expression stability of eight commonly used RGs (*TBP*, *GUSB*, *POLR2A*, *PPIA*, *B2M*, *GAPDH*, *ACTB*, and *HPRT1*) for their use in gene expression analysis with RT–qPCR. We compared the stability of such RGs in PBMCs using four different algorithms. To our knowledge, this is the first study to validate RG stability in PBMCs treated with DEX and Vit D_3_ in obese asthmatic patients compared to non-obese patients with and without asthma. Only a few studies have tackled this issue [8,23,24], and, to the best of our knowledge, none have compared RG stability in these diseases. 

We found that all analyzed genes were expressed in PBMCs from healthy, asthmatic, and obese asthmatic subjects. *B2M* was the most expressed gene, and *HPRT1* was the least expressed gene. These results are in agreement with those of Oturai et al., who reported a low expression of *HPRT1* in PBMCs from healthy and multiple sclerosis patients [25]. In this study, the most variable gene was *GAPDH,* despite being a frequently used RG. This observation is in line with previous studies reporting high variability for this RG in PBMCs [25,26].

We found differences in RG expression between healthy, asthmatic, and obese asthmatic patients, and we suggest that these differences could have been due to different experimental conditions such as distinct incubation times and/or distinct in vitro treatments. It is known that PBMC gene expression could be modified as a function of waiting time in sample processing or as a function of cell culture incubation time [26]. We found no differences between 6 and 24 h incubations in PBMCs from healthy individuals. However, *GAPDH* presented differences in both asthmatic and obese asthmatic groups, and *PPIA* and *TBP* presented differences in asthmatic PBMCs. Moreover, RG expression could change according to their stimulation [18]. We assessed the impact of in vitro GC and vitamin D treatment on RG stability. PBMCs were treated with DEX with or without Vit D_3_ for 6 and 24 h. *POLR2A* was the least stable gene, being expressed differently in PBMCs from asthmatic and obese asthmatic patients treated with DEX, Vit D_3,_ or both treatments combined. In contrast, the *B2M*, *PPIA*, and *ACTB* genes were not affected by the in vitro treatments.

Altogether, these findings suggest that not all RGs retain stability under distinct culture times and in vitro treatments, thus evidencing the importance of measuring their stability under the specific conditions of each study, such as the study population and experimental conditions.

We found that all analyzed RGs could be considered stable according to stability values calculated via the distinct algorithms, although there were some discrepancies according to the patients’ characteristics. These different results depending on the disease are in accordance with the work of Usarek et al., who found that the best RG for healthy PBMCs was *B2M*, whereas in PBMCs from patients with amyotrophic lateral sclerosis, it was *HPRT1* [27]. Additionally, Wang et al. found lower basal expression levels of *PPIA* in asthmatics than in healthy controls [15]. The cell type used for the study is also important, as the stability of the possible RGs may be affected. In our case, we used PBMCs, which are a mixture of different cell types, so there may be variability in the stability of RG expression, as others have shown in their studies [22,25].

Consistent with the MIQE (minimum information for publications of quantitative real-time PCR experiments) guidelines, our results support a normalization strategy based on at least two RGs [9], with three genes being the optimal number for healthy and obese asthmatic PBMCs and four genes being the optimal number for asthmatic samples. Furthermore, with the NormFinder algorithm, we investigated whether the stability of RGs was subject to variation according to each experimental treatment. *B2M* was the gene with the least intergroup variation in healthy and obese asthmatic PBMCs; however, *HPRT1* presented the least intergroup variation in the asthmatic samples.

To date, there is no consensus on what method should be used to study RG expression stability, and, because the programs are based on different algorithms, the consensus between them increases the reliability of the results. Unfortunately, we did not identify any RG that was universally stable across these four programs. To overcome the discrepancies and obtain a final ranking, we calculated a comprehensive ranking based on the geometric mean of the positions for each program.

Finally, we identified the best and worst shared genes between the three studied groups. *B2M* and *HPRT1* were the most stable RGs in PBMCs from healthy, asthmatic, and obese asthmatic patients. These results contrast with those of Nakayama et al., who found *B2M* to be an unstable gene for chronic rhinosinusitis patients [28], and Ledderose et al., who stated that *TBP* was the best RG for T cell and neutrophil gene expression analysis [19]. On the other hand, *GAPDH* and *TBP* were the worst studied RGs, which is in line with previous studies that described *GAPDH* as an unstable gene along with *ACTB*, which was also positioned last in our ranking [29,30]. 

Since obesity is considered to influence the disease process of asthma, exploring the underlying mechanisms is necessary. Therefore, genetic expression assays could serve as a good in vitro model to study GC resistance in PBMCs. However, the selection of suitable RGs is the first and crucial step before we can determine the molecular basis. We identified *HPRT1* and *B2M* as the most stable RGs and *ACTB*, *TBP*, and *GAPDH* as the least stable RGs. This screening is a preliminary analysis for future gene expression studies in PBMCs from obese asthmatic patients treated in vitro with DEX and/or Vit D_3_.

## Figures and Tables

**Figure 1 mps-05-00035-f001:**
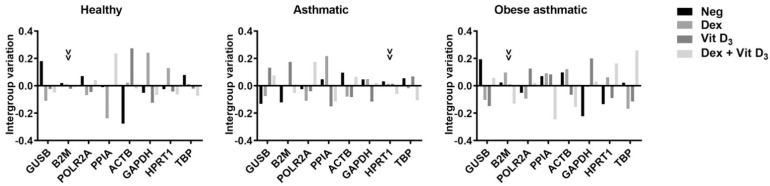
Intergroup variations in PBMCs from healthy subjects, asthmatic subjects, and obese asthmatic subjects. Arrowheads point to the most stable gene with the least intergroup variation. Neg: negative control; Dex: dexamethasone; Vit D_3_: 1,25-dihydroxyvitamin D.

**Figure 2 mps-05-00035-f002:**
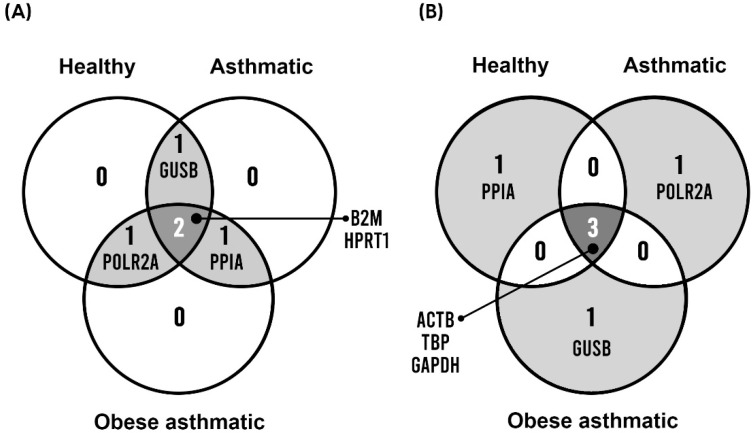
Best (**A**) and worst (**B**) shared genes among healthy, asthmatic, and obese asthmatic groups.

**Table 1 mps-05-00035-t001:** Ranking of potential RGs according to stability values calculated with different algorithms.

Group	Reference Genes	Stability Value					Comprehensive Ranking
		BestKeeper		ΔCt	geNorm	NormFinder	
		SD	r	SD	M	S	
Healthy	*HPRT1*	0.50	0.89	0.58	0.58	0.08	1.32
	*GUSB*	0.45	0.76	0.63	0.63	0.11	2.06
	*B2M*	0.54	0.83	0.63	0.63	0.11	2.63
	*POLR2A*	0.47	0.75	0.64	0.64	0.12	3.36
	*ACTB*	0.55	0.76	0.69	0.69	0.15	5.48
	*TBP*	0.75	0.89	0.68	0.68	0.16	5.69
	*PPIA*	0.67	0.74	0.78	0.78	0.19	6.74
	*GAPDH*	0.85	0.82	0.87	0.87	0.19	8.00
Asthmatic	*GUSB*	0.76	0.91	0.67	0.67	0.09	1.32
	*HPRT1*	0.80	0.91	0.70	0.70	0.13	3.16
	*PPIA*	0.75	0.90	0.72	0.72	0.22	3.46
	*B2M*	0.64	0.69	0.91	0.91	0.13	3.72
	*TBP*	0.78	0.84	0.76	0.76	0.13	4.47
	*POLR2A*	0.82	0.89	0.72	0.73	0.15	4.90
	*GAPDH*	1.00	0.92	0.81	0.81	0.12	5.29
	*ACTB*	0.84	0.91	0.77	0.77	0.18	6.48
Obese Asthmatic	*B2M*	0.59	0.91	0.61	0.61	0.10	1.19
	*PPIA*	0.57	0.86	0.66	0.66	0.15	2.11
	*POLR2A*	0.60	0.86	0.66	0.66	0.15	3.57
	*HPRT1*	0.75	0.92	0.66	0.66	0.13	3.87
	*TBP*	0.65	0.85	0.70	0.70	0.13	4.16
	*ACTB*	0.73	0.91	0.70	0.70	0.14	5.42
	*GUSB*	0.71	0.79	0.76	0.76	0.20	6.44
	*GAPDH*	0.77	0.82	0.79	0.79	0.20	8.00

SD: standard deviation; r: Pearson correlation coefficient; M: geNorm stability value; S: NormFinder stability value.

## Data Availability

Data generated or analyzed during this study are included in this published article or available from the corresponding author on request. All analyzed datasets were sourced by the authors.

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
