# Peer review of "Reference Gene Validation for RT–qPCR in PBMCs from Asthmatic Patients with or without Obesity"

_mps, 2022, doi:10.3390/mps5030035_

Round 1

Reviewer 1 Report

This manuscript entitled “Reference gene validation for RT-qPCR in PBMCs from asthmatic patients with or without obesity “ by Bantula et al analyzes reference genes used in RT-qPCR in PBMC RNAs in obesity-associated asthma studies. This work provides important information for the use of appropriate reference genes that can be used by researchers in studies on obesity-associated asthma. The following are comments meant for clarification to improve the manuscript.

Comments:

  1. For RNA extraction, what were the methods used to ensure that genomic DNA was removed from the final extracted nucleic acids?
  2. This paper has N=3 biological replicates. For the cell culture and RT-PCR per experimental condition, the authors used only one representative sample for every biological sample which was surprising. Could this have led to the apparent intergroup variations in the results? How many technical repeats were included in the real-time PCR? It is important to include this information in validation reports such as this type of manuscript.
  3. What was the rationale for the concentrations of Dex and Vitamin D3 used in the study? It would be interesting to see if there are changes in the reference genes tested using different concentrations of Dex and Vitamin D3

Reviewer 2 Report

The author’s reference gene (GUSB, B2M, POLR2A, PPIA, ACTB, GAPDH, HPRT1, TBP) validation for RT-qPCR in PBMCs from asthmatic patients with or without obesity is a very good and useful study. The stability was analyzed using four different algorithms— BestKeeper, ΔCt, geNorm, and NormFinder. GeNorm analysis and the results showed B2M and HPRT1 were the most stable RGs among all groups, whereas ACTB, TBP, and GAPDH were the worst-shared ones.

Overall, the authors have done impressive work in this review. This review is highly relevant to the field of obesity-related diseases.

I have only one comment.

Supplementary Table S3: Quantification cycle values in healthy, asthmatic, and obese asthmatic PBMCs, could you please specify the exact P-value with respect to each group comparison.?
